# Serum Erythroferrone Levels Associate with Mortality and Cardiovascular Events in Hemodialysis and in CKD Patients: A Two Cohorts Study

**DOI:** 10.3390/jcm8040523

**Published:** 2019-04-16

**Authors:** Belinda Spoto, Rahul Kakkar, Larry Lo, Matt Devalaraja, Patrizia Pizzini, Claudia Torino, Daniela Leonardis, Sebastiano Cutrupi, Giovanni Tripepi, Francesca Mallamaci, Carmine Zoccali

**Affiliations:** 1CNR-IFC, Clinical Epidemiology of Renal Diseases and Hypertension, Reggio Cal Unit, 89126 Pisa, Italy; belinda.spoto@tin.it (B.S.); ppizzini@ifc.cnr.it (P.P.); ctorino@ifc.cnr.it (C.T.); dleonardis@ifc.cnr.it (D.L.); scutrupi@ifc.cnr.it (S.C.); gtripepi@ifc.cnr.it (G.T.); francesca.mallamaci@libero.it (F.M.); 2Corvidia, Waltham, MA 02451, USA; rkakkar@corvidiatx.com (R.K.); llo@corvidiatx.com (L.L.); mdevalraja@corvidiatx.com (M.D.)

**Keywords:** erythroferrone, chronic kidney disease, dialysis, mortality, iron

## Abstract

Erythroferrone (ERFE) is a hepcidin inhibitor whose synthesis is stimulated by erythropoietin, which increases iron absorption and mobilization. We studied the association between serum ERFE and mortality and non-fatal cardiovascular (CV) events in a cohort of 1123 hemodialysis patients and in a cohort of 745 stage 1–5 chronic kidney disease (CKD) patients. Erythroferrone was measured by a validated enzyme-linked immunosorbent assay (ELISA). In the hemodialysis cohort, serum ERFE associated directly with erythropoiesis stimulating agents (ESA) dose (*p* < 0.001) and inversely with serum iron and ferritin (*p* < 0.001). Erythroferrone associated with the combined outcome in an analysis adjusting for traditional risk factors, factors peculiar to end-stage kidney disease, serum ferritin, inflammation, and nutritional status (HR, hazard ratio, (5 ng/mL increase: 1.04, 95% confidence interval, CI: 1.01–1.08, *p* = 0.005). Furthermore, treatment with ESA modified the relationship between ERFE and the combined end-point in adjusted analyses (*p* for the effect modification = 0.018). Similarly, in CKD patients there was a linear increase in the risk for the same outcome in adjusted analyses (HR (2 ng/mL increase): 1.04, 95% CI: 1.0–1.07, *p* = 0.015). Serum ERFE is associated with mortality and CV events in CKD and in HD patients, and treatment by ESA amplifies the risk for this combined end-point in HD patients.

## 1. Introduction

Iron plays a crucial role in several fundamental biological processes such as oxygen transport, cellular respiration, and metabolic reactions [1]. A complex system of proteins and hormones controls iron metabolism, and hepcidin is a major regulator of this system [2]. This protein reduces the intestinal absorption of dietary iron and the release of stored iron from hepatocytes and macrophages, a process involving the cellular iron exporter ferroportin. Iron and inflammation are both hepcidin enhancers, while hypoxia-inducible factors, the sex hormones estrogen and testosterone, downregulate hepcidin [2]. In 2014, a new iron metabolism regulating factor synthesized in erythroblasts in response to erythropoietin—erythroferrone (ERFE)—was identified [3]. This hormone is encoded by the *FAM132B* gene—now renamed ERFE gene—and coincides with a protein also expressed in the skeletal muscle, called myonectin (CTRP15) [4]. By suppressing hepcidin, ERFE increases the absorption and mobilization of iron to provide an adequate iron supply during stress erythropoiesis such as during rapid growth or blood loss [3]. Interest in ERFE in human diseases goes beyond anemia because this factor is synthesized also in muscle cells (myonectin), and there is emerging evidence that this factor may help to prevent ischemia-reperfusion injury [5,6].

Anemia is a hallmark of chronic kidney disease (CKD) [7], and erythropoietin (EPO) deficiency and reduced iron bioavailability by high hepcidin levels are fundamental factors underlying this condition in CKD [8]. Biomarkers of iron metabolism like serum ferritin and hepcidin have been independently associated with the risk of death [9,10] and cardiovascular (CV) events [11] in this population. Information on ERFE in CKD is limited to just two studies [12,13], and in both studies circulating ERFE showed a dose-response relationship with the amount of erythropoiesis stimulating agents (ESA) administered, which is in line with experimental knowledge in animal models [3]. Because of the peculiar combination of risk factors underlying anemia [14] and the risk for CV events [15,16] in CKD, studying the relationship between ERFE, mortality, and CV disease is a relevant issue that has never been investigated in this population.

With this background in mind, we studied the relationship between serum ERFE, mortality, and CV events in a cohort of predialysis patients with CKD of various severities and in a separate cohort of end-stage kidney disease (ESKD) patients maintained on chronic hemodialysis (HD). Because of the physiological relationship between serum ERFE and EPO levels, a second pre-specified goal of this study was to test whether the relationship of ERFE with mortality and non-fatal CV events is modified by the dose of ESA in these patients.

## 2. Methods

### 2.1. Study Protocol

The protocol confored to the ethical guidelines of our institution and was in adherence with the Declaration of Helsinki. Informed written consent was obtained by each participant. 

### 2.2. Patients

The present study included two independent cohorts: a cohort of patients on hemodialysis treatment (HD cohort) and a cohort of patients with CKD of various severities (CKD cohort).

#### 2.2.1. Hemodialysis Cohort

The original study cohort included 1189 Caucasian hemodialysis (HD) patients enrolled from February 2009 to October 2010 in two regions (Calabria and Sicily) in Southern Italy (the PROspective Registry of the Working GRoup of Epidemiology of DIalysis REgion Calabria, PROGREDIRE [17]). Patients in this cohort represented about the 80% of the whole HD population in the geographical area covered by the 35 dialysis units participating into the study. They had been on regular HD treatment for a median time of 47 months (inter-quartile range, IQR: 22−90 months) and were being treated thrice weekly with standard bicarbonate dialysis with non-cellulosic membrane filters of various type. From the source population of 1189 patients, 65 patients were excluded because of missing blood samples and one patient because of extremely high serum level of ERFE (560.45 ng/mL) identified as a statistically significant outlier by the Grubbs’ test (*p* < 0.001). Thus, 1123 patients were considered for the data analysis. The individual dose of ESA in this cohort was computed by adopting the conversion dose darbepoetin α 1µg = 200 units epoietin α. All HD patients received intravenous iron (iron gluconate) along the recommendations of KDOQI guidelines at the time of the study [18]. Iron therapy was profiled according to the clinical response. To prevent iron overload, intravenous iron was temporary stopped when ferritin levels were >500 ng/mL.

#### 2.2.2. Chronic Kidney Disease Cohort

This population included 745 out of 759 patients (in 14 patients, ERFE measurement could not be performed because of missing blood samples) with CKD of various severity (stages 1–5) recruited from 22 Nephrology Units in Southern Italy between October 2005 and September 2008 (the MAURO study cohort [19]). All patients were in stable clinical condition, and none had acute or rapidly evolving renal diseases, intercurrent infections, or acute inflammatory processes. These patients were also non-transplanted, non-pregnant, and not affected by cancer or diseases in the terminal phase. CKD patients not on dialysis were treated with oral iron supplements if they had laboratory evidence of iron deficiency.

### 2.3. Study Outcome

The outcome of the study was a combined end-point composed by all-cause mortality and non-fatal CV events. CV events included myocardial infarction as confirmed by serial changes of electrocardiographic and cardiac biomarkers, electrocardiographically documented angina episodes, electrocardiographically documented serious arrhythmia (atrial flutter or fibrillation, ventricular fibrillation), and de novo chronic heart failure. To be classified as having chronic heart failure, patients had to show mild or more severe dyspnea during ordinary activities (New York Heart Association class II or higher) plus evidence of anatomical/functional left ventricular disease on echocardiography; stroke documented by computed tomography, magnetic resonance imaging, and/or clinical and neurological evaluation; transient ischemic attacks; or unexpected, sudden death highly suspected to be of cardiac origin. Causes of death were assessed by three independent physicians. In doubtful cases, diagnosis was attributed by consensus. During the review process, the involved physicians used all available medical information, including hospitalization forms and medical records. In the case of death occurring at home, family members and/or general practitioners were interviewed to better understand the circumstances that led to death.

### 2.4. Laboratory Measurements

In both cohorts, blood sampling was performed in the early morning after an overnight fast and during a mid-week non-dialysis day for HD patients. Blood was collected in EDTA-containing tubes, and serum/plasma supernatants were stored at −80 °C until batch analyses. Serum glucose, cholesterol, hemoglobin, ferritin, iron, albumin, phosphate, parathyroid hormone (PTH), creatinine, and high sensitivity C-reactive protein (CRP) were made using standard methods in the routine clinical laboratory. Estimated glomerular filtration rate (eGFR) was calculated by using the four-variables MDRD study equation. All CKD patients underwent a 24 h urinary collection for the measurement of proteinuria.

Serum ERFE was determined by enzyme-linked immunosorbent assay (Intrinsic LifeSciences, La Jolla, CA, USA) based on the well validated method by Ganz et al. [20].

### 2.5. Statistical Analysis

Data were expressed as mean ± standard deviation for normally distributed data, median and inter-quartile range (IQR) for non-normally distributed data, or as percent frequency. Variables were compared by t-test or Mann-Whitney U test, as appropriate. The distribution of serum ERFE levels was positively skewed, and the kit used to measure this biomarker was unable to detect ERFE values <1.60 ng/mL. Therefore, a theoretical value of 1.59 ng/mL was attributed to all patients with undetectable values of ERFE. Erythroferrone was unmeasurable (ERFE < 1.60 ng/mL) in 26% of patients of the HD cohort and in 49% of patients of the CKD cohort. To minimize the potential analytical distortion due to the replacement of unmeasurable ERFE levels with the value of 1.59 ng/mL, a sensitivity analysis excluding patients with undetectable ERFE levels was performed in both cohorts. The correlation analyses between continuous variables were performed by the Pearson’s product moment correlation coefficient (r) or Spearman rank correlation coefficient (rho), while the correlation analysis between continuous and binary variables was done by the point-biserial correlation coefficient. Variables having a positively skewed distribution were log transformed (ln) before the correlation study.

Survival analysis aimed at investigating the relationship between serum ERFE (independent variable) and the combined outcome death or incident non-fatal CV events (dependent variable) was performed by Cox regression analysis. In the HD cohort, multivariate Cox model included serum ERFE as a continuous variable as well as traditional risk factors (age, sex, current smoking, diabetes, total cholesterol, blood pressure, and background CV events), factors peculiar to ESKD (phosphate, hemoglobin, dialysis vintage, Kt/V, and PTH), biomarkers of iron metabolism (serum ferritin and iron), and biomarkers of inflammation and nutritional status (CRP, albumin and body mass index (BMI)). In the CKD cohort, a parsimonious model based on backwardly selected variables (i.e., age, sex, background CV events, albumin, and hemoglobin) among a large list of variables (those included in Table 1) was adopted because of the relatively small number of events in this cohort. The proportionality assumption was tested by analyzing the Schoenfeld residuals, and no violation was found. The interaction analysis was performed by the standard linear combination method. The potential effect modification by ESA dose on the relationship between serum ERFE and the combined outcome was analyzed by creating appropriate multiplicative terms in Cox regression analyses. Data were expressed as hazard ratios (HR), 95% confidence intervals (CI), and *p* values. All statistical analyses were performed by using standard statistical packages (SPSS for Windows, Version 24; SPSS Inc., Chicago, Illinois, USA; STATA for Windows, Version 13; Stata Corp., College Station, Texas, USA).

## 3. Results

The main demographic, somatometric, clinical, and biochemical characteristics of the two study populations are reported in Table 1.

In the HD cohort, 64% of patients were males and had a mean age of 65 ± 14 years. Three hundred and one patients (27%) were affected by diabetes, 157 (14%) were habitual smokers, and 586 (52%) had CV comorbidities. The dialysis vintage was 47 months (IQR: 22–90), and the Kt/V was 1.28 ± 0.41. Seven hundred and eighty-five patients (70%) were on ESA treatment, and 349 (31%) were treated with iron compounds. The primary renal disease was hypertensive nephropathy in 335 (30%), glomerulonephritis in 145 (13%), pyelonephritis in 85 (8%), autosomal dominant polycystic renal disease or other genetic diseases in 134 (12%), diabetic nephropathy in 184 (16%), other causes in 26 (2%), and unknown in 214 (19%).

In the CKD cohort, patients had a mean age of 62 ± 11 years, 297 (40%) were males, 260 (35%) patients had diabetes, 372 (50%) were current smokers, and 217 (29%) had a history of CV disease. The mean eGFR was 36 ± 13 mg/mL/1.73m^2^, and the median proteinuria was 0.6 g/24 hours (IQR: 0.21–1.49 g/24 hours). Only 66 (9%) patients were treated with ESA, and 57 (8%) received iron compounds. Among the CKD stages, patients were distributed as follows: CKD stage G1, 3%; stage G2, 22%; stage G3, 38%; stage G4, 34%; and stage G5, 3%. The primary cause of CKD was glomerulonephritis in 64 (9%) patients, diabetic nephropathy in 61 (8%), interstitial nephropathy or pyelonephritis in 34 (4.5%), hypertensive nephropathy in 85 (11%), autosomal dominant polycystic renal disease or other genetic diseases in 62 (8.5%), other causes in 46 (6%), and unknown in 393 (53%) patients.

### 3.1. Clinical Correlates of Serum Erythroferrone

In both cohorts, serum levels of ERFE had a left skewed distribution with a median value of 4.5 ng/mL and a wide range (from 1.6 to 140.4 ng/mL; mean value of 7.8 ±10.5 ng/mL) in HD patients, and a median of 1.6 ng/mL and a similarly wide range (from 1.6 to 127.8 ng/mL; mean of 3.4 ± 7.5 ng/mL) in CKD patients with no difference among sexes (HD cohort (males: median 4.47 ng/mL and females: median 4.69 ng/mL; *p* = 0.29); CKD cohort (males: median 1.65 ng/mL and females: median 1.62 ng/mL; *p* = 0.92)) (Appendix A).

In HD patients, log-transformed ERFE was directly related to age (r = 0.13, *p* < 0.001), dialysis vintage (r = 0.12, *p* < 0.001), and PTH (r = 0.06, *p* = 0.08), and inversely associated with iron (r = −0.17, *p* < 0.001), hemoglobin (r = −0.16, *p* < 0.001), albumin (r = −0.13, *p* < 0.001), ferritin (r = −0.12, *p* < 0.001), cholesterol (r = −0.08, *p* = 0.01), and systolic blood pressure (r = −0.08, *p* = 0.01) but independent of BMI, diabetes, smoking, background CV comorbidities, phosphate, CRP, and Kt/V (P ranging from 0.11 to 0.98). Furthermore, ERFE was associated with the dose of ESA both on Spearman regression analysis (ρ = 0.16, *p* < 0.001) and in a categorical analysis based on 50th percentile of the ERFE distribution *(p <* 0.001) (Figure 1).

In the CKD cohort, log-transformed serum ERFE was directly related to PTH (r = 0.13, *p* = 0.002), age (r = 0.11, *p* = 0.002), CRP (r = 0.10, *p* = 0.009), and background CV comorbidities (r = 0.09, *p* = 0.02) and inversely related to iron (r = −0.18, *p* < 0.001), hemoglobin (r = −0.12, *p* = 0.002), and cholesterol (r = −0.11, *p* = 0.003). Serum ERFE in this cohort was unrelated to GFR (*p* = 0.50) and serum ferritin (*p* = 0.87). Additionally, in this cohort, serum ERFE and ESA treatment were strongly associated, with this biomarker being significantly higher (*p* < 0.001) in ESA-treated patients (2.71 ng/mL, IQR: 1.71–5.85 ng/mL) than in those untreated with this drug (1.6 ng/mL, IQR: 1.59–2.66 ng/mL).

### 3.2. Serum Erythroferrone, Mortality, and Non-Fatal Cardiovascular Events in Hemodyalisis Patients 

During a median follow-up of 27 months (IQR: 13 to 43 months), 627 patients died or had non-fatal CV events. The causes of death are reported in Appendix A. Serum ERFE associated with the risk of mortality and non-fatal CV events in crude analyses considering serum ERFE as a continuous variable in its original form (HR (5 ng/mL increase): 1.05, 95% CI: 1.03–1.08, *p* < 0.001)) or as a log-transformed variable (HR (1 ln unit increase): 1.26, 95% CI: 1.16–1.36, *p* < 0.001) and in adjusted analyses (Table 2). A sensitivity analysis excluding patients with undetectable levels of ERFE (≤1.59 ng/mL) carried out in 830 HD patients confirmed this association (Appendix A).

### 3.3. Serum Erythroferrone, Mortality, and Non-Fatal Cardiovascular Events in Chronic Kidney Disease Patients

One hundred and twenty-six out of 745 CKD patients died or experienced non-fatal CV events during the follow-up period (median: 36 months, IQR: 30 to 36 months). On univariate Cox regression analysis, a 2 ng/mL higher serum ERFE signaled a 3% excess risk of mortality and non-fatal CV events (HR: 1.03, 95% CI: 1.01–1.06, *p* = 0.012). Similarly, serum ERFE was related to the combined outcome also after log-transformation (HR (1 ln unit increase): 1.46, 95% CI: 1.19–1.79, *p* < 0.001). This association remained substantially unchanged after adjustment for age, sex, albumin, hemoglobin, phosphate, and background CV events (the covariates selected with a backward approach, see Methods) (Table 3). A sensitivity analysis excluding patients on ESA treatment confirmed the association of serum ERFE with mortality and non-fatal CV events on a multivariate Cox regression analysis adjusting for age, smoke, albumin, phosphate, hemoglobin, PTH, and background CV events (Appendix A). In a further analysis restricted to patients with detectable levels (>1.59 ng/mL) of ERFE (n = 304), the association remained strong and significant (Appendix A).

### 3.4. Serum Erythroferrone and Combined Outcome: Effect Modification by ESA Treatment in the Hemodyalisis Cohort

Treatment with ESA was a modifier of the relationship between serum ERFE and the combined outcome both on univariate (*p* for the effect modification = 0.028) and multivariate analyses adjusting for a large set of potential confounders including age, sex, current smoking, diabetes, total cholesterol, blood pressure and background CV events, factors peculiar to ESKD (phosphate, hemoglobin, dialysis vintage, Kt/V, PTH), biomarkers of iron metabolism (serum ferritin and iron), and biomarkers of inflammation and nutritional status (CRP, albumin and body mass index (BMI) (*p* for the effect modification = 0.018). As shown in Figure 2, the risk excess for mortality and non-fatal CV events by ERFE was progressively higher across increasing doses of ESA treatment. Of note, this interaction was specific because no similar effect modification existed for other traditional or non-traditional risk factors listed in Table 1.

## 4. Discussion

In this study, ERFE was coherently associated with mortality and CV events in two separate cohorts with CKD of various severity. Serum ERFE was dependent on ESA treatment, and this treatment amplified the risk for the study outcome by ERFE. 

Erythroferrone, a well characterized hepcidin suppressor, is key to ensuring an adequate iron supply when erythropoiesis is stimulated [21]. Knowledge on this hormone in CKD is limited to two studies. In the first, serum ERFE levels in HD patients were substantially similar to those in control subjects and 10 times lower than the levels in HD patients in the present study (0.5 ng/mL vs. 4.5 ng/mL) [12]. In the second study by Hanudel et al. [13], serum ERFE levels, measured by the recent assay by Ganz et al. [20], were similar in 51 CKD and 161 healthy control subjects (6.1 (2.6–15.0) ng/mL and 7.8 (4.7–13.2) ng/mL, respectively) but twice lower than in 97 HD patients (15.7 (7.9–32.5) ng/mL). In the present study, including 1123 HD patients and 745 CKD patients, we measured serum ERFE by a kit based on Ganz assay and found that ERFE was much higher in HD patients than in CKD patients but, on average, ERFE levels in our patients were lower than in the corresponding patients’ groups in Hanudel study [13].

In this study, ERFE emerged as a coherent direct risk factor of death and CV complications in two separate CKD cohorts. Erythroferrone is strongly related in an inverse fashion to hepcidin in HD patients [13], which is per sé a direct correlate of CV events in HD patients [11]. Therefore, our observation that ERFE is directly associated with the combined outcome is prima facie counterintuitive with this finding. Furthermore, the relationship between ERFE and the study outcomes was largely independent of major inflammation markers like CRP, serum iron, and ferritin, suggesting that interference with these factors does not explain the excess risk for death and CV events by relatively higher levels of ERFE in CKD and in HD patients. It is important to note that ERFE synthesized in the skeletal muscle in response to exercise (myonectin) has a protective role for the CV system in experimental models [5]. Even though discrepancies among myonectin and ERFE assays [4,12,20] still need to be understood and reconciled, it should be noted that other biomarkers that per sé underlie a protective action for the CV system, like adiponectin—a paralogue to ERFE—are directly, rather than inversely, related to the risk of death in HD patients [22]. Likewise, brain natriuretic peptide, another hormone endowed with cardio-protective actions, is a direct predictor of death and CV events in CKD [23] and in ESKD [24] patients as well. Alternatively, it is possible that the effect of ERFE on the CV system is more complex than that is currently known or has a different effect in men vs. mice. 

Of note, in the HD cohort, treatment with ESA was not only associated with higher levels of ERFE but also modified the relationship between this factor and the study outcome. This interaction was notable because no effect modification by other variables (those listed in Table 1) on the ERFE-combined outcome relationship was observed. ESA treatment targeting a higher vs. a lower hemoglobin level is associated with a higher risk of stroke, hypertension, and vascular access thrombosis in CKD patients [25], a phenomenon possibly attributable to increased cytoplasmic Ca^2+^ concentrations and increased endothelin-1 production in a context of reduced nitric oxide bioavailability [26] like in CKD [27]. In light of the fact that myonectin (ERFE) induced by treadmill exercise mitigates the inflammatory response and limits the infarcted area by ischemia-reperfusion injury in mice [6], it can be speculated that ERFE is part of a regulatory response triggered by the adverse effects of ESA treatment on the vascular system.

Our study has limitations. First, our cohorts included patients of Caucasian descent only. Therefore, it remains to be seen whether findings in this study can be generalized to other ethnicities. Second, we did not measure the target molecule of ERFE—hepcidin. Hepcidin is in the pathogenic pathway whereby ERFE impacts upon iron metabolism and anemia and, ultimately, on clinical outcomes. However, adjustment for inflammation, the main driver of hepcidin in CKD, and for a main indicator of iron stores like ferritin had minimal, if any, effect on the risk for the combined outcome by ERFE in both HD and CKD cohorts included in this study. Third, our findings can merely suggest, but not prove, causality. Our interpretation that ERFE is released as part of a biological response aimed at countering the high risk of CKD needs to be tested in appropriate experimental studies.

## 5. Conclusions

In conclusion, serum ERFE is a risk factor of mortality and CV events in HD and CKD patients and treatment by ESA amplifies the risk for this combined end point. These observations suggest that ERFE may be part of the biological response to a high risk condition like CKD, a hypothesis that remains to be tested in mechanistic studies.

## Figures and Tables

**Figure 1 jcm-08-00523-f001:**
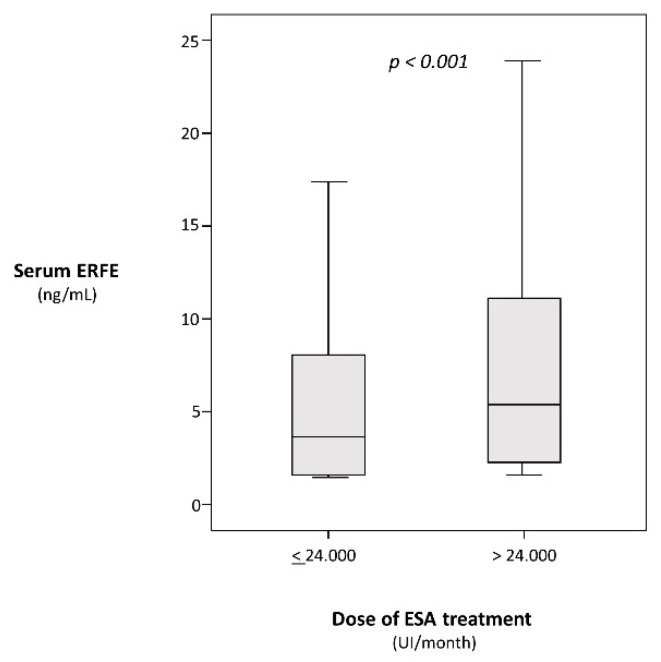
Serum erythroferrone (ERFE) levels below and above the median value (24.000 UI) of monthly erythropoiesis stimulating agents (ESA) dose.

**Figure 2 jcm-08-00523-f002:**
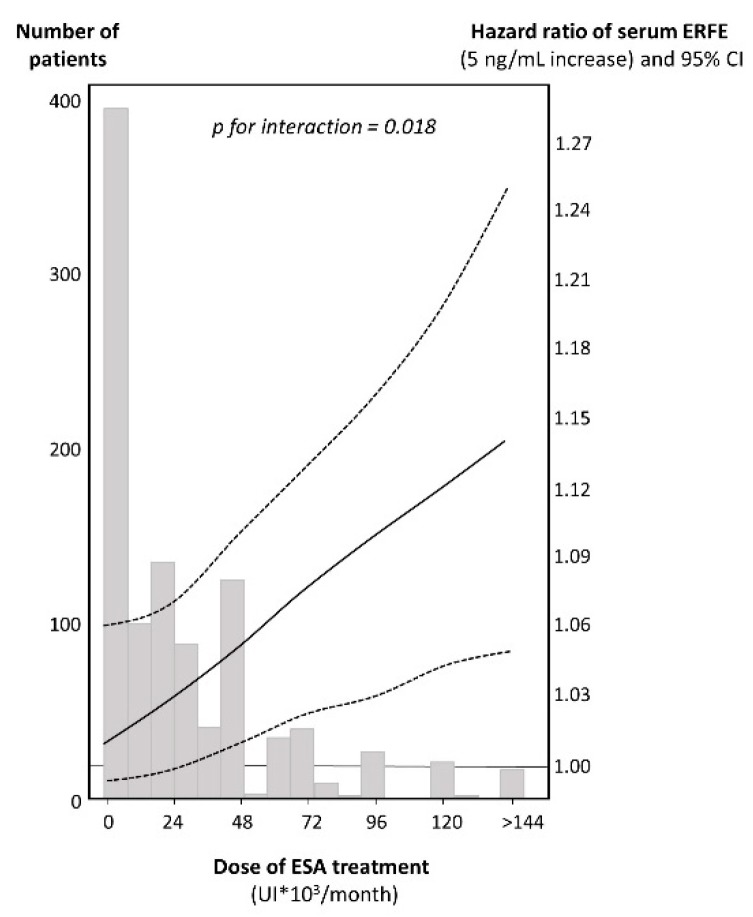
Interaction between ESA dose and serum ERFE levels for the risk of death and non-fatal CV events in HD patients. The hazard ratio for the combined end-point portended by a fixed increase (5 ng/ml) in serum ERFE levels is reported on the right scale. The continuous line represents the shape of the hazard ratios throughout the doses of ESA treatment and the dotted lines the corresponding 95% CI. In the background, the distribution of ESA doses is plotted, and the number of patients corresponding to each column of the histogram is reported on the left scale. Data are adjusted for age, sex, smoke, diabetes, cholesterol, blood pressure, background CV events, factors peculiar to ESKD (phosphate, hemoglobin, dialysis vintage, Kt/V, PTH), biomarkers of iron metabolism (serum ferritin and iron), and biomarkers of inflammation and nutritional status (CRP, albumin and BMI).

**Table 1 jcm-08-00523-t001:** Main demographic, somatometric, clinical, and biochemical characteristics of the two study populations.

	HD Cohort (*n* = 1123)	CKD Cohort (*n* = 745)
Age, years	65 ± 14	62 ± 11
Male sex, n (%)	720 (64)	297 (40)
Smokers, n (%)	157 (14)	372 (50)
Diabetic, n (%)	301 (27)	260 (35)
Cholesterol, mg/dL	154 ± 39	187 ± 45
Systolic blood pressure, mmHg	135 ± 22	134 ± 18
BMI, kg/m^2^	25.1 ± 4.7	28.2 ± 4.7
With background CV events, n (%)	586 (52)	217 (29)
On ESA treatment, n (%)	785 (70)	66 (9)
On iron treatment, n (%)	349 (31)	57 (8)
Albumin, g/dL	3.9 ± 0.5	4.2 ± 0.5
Phosphate, mg/dL	5.0 ± 1.6	3.7 ± 0.8
Hemoglobin, g/dL	11.3 ± 1.5	12.8 ± 1.8
Ferritin, ng/mL	255 (104–591)	83 (39–155)
Iron, μg%	68.6 ± 34.6	72.8 ± 28.3
C-reactive protein, mg/L	4.5 (2.9–12.0)	2.4 (1.0–5.5)
Parathyroid hormone, pg/mL	243 (118–460)	76 (52–125)
Glomerular filtration rate, ml/min/1.73m^2^		36 ± 13
Urinary protein, g/24h		0.60 (0.21–1.49)
Dialysis vintage, mth	47 (22–90)	
Kt/V	1.28 ± 0.41	

Data are expressed as mean ± standard deviation, as median and inter-quartile range, or as percent frequency, as appropriate.

**Table 2 jcm-08-00523-t002:** Adjusted association between serum ERFE and mortality and non-fatal cardiovascular (CV) events in the chronic hemodialysis (HD) cohort.

	Unit of Increase	Hazard Ratio (95% CI), *p* Value
ERFE	5 ng/mL	1.04 (1.01−1.08), *p* = 0.005
Age	1 year	1.04 (1.03−1.04), *p* < 0.001
Male sex		1.15 (0.96−1.36), *p* = 0.13
Smoking	yes/no	1.07 (0.84−1.37), *p* = 0.57
Diabetes	yes/no	1.33 (1.10−1.60), *p* = 0.003
Cholesterol	1 mg/dL	0.99 (0.99−1.01), *p* = 0.09
Systolic blood pressure	1 mmHg	1.01 (0.99−1.01), *p* = 0.09
Background CV events	yes/no	1.50 (1.26−1.77), *p* < 0.001
Hemoglobin	1 g/dL	1.02 (0.96−1.08), *p* = 0.52
C-reactive protein	25 mg/L	0.99 (0.95−1.04), *p* = 0.83
Ferritin	50 ng/mL	1.01 (1.01−1.02), *p* = 0.003
Iron	1 μg%	0.99 (0.99−1.01), *p* = 0.09
BMI	1 kg/m^2^	0.99 (0.98−1.02), *p* = 0.70
Albumin	1 g/dL	0.65 (0.53−0.79), *p* < 0.001
Phosphate	1 mg/dL	1.06 (1.01−1.12), *p* = 0.04
Parathyroid hormone	50 pg/mL	1.01 (0.99−1.02), *p* = 0.06
Dialysis vintage	1 month	1.01 (1.00−1.01), *p* = 0.001
Kt/V	1 unit	0.99 (0.79−1.26), *p* = 0.95

ERFE: erythroferrone.

**Table 3 jcm-08-00523-t003:** Adjusted association between serum ERFE and mortality and non-fatal CV events in the CKD cohort.

	Unit of Increase	Hazard Ratio (95% CI), *p* Value
ERFE	2 ng/mL	1.04 (1.01–1.07), *p* = 0.015
Age	1 year	1.07 (1.05–1.10), *p* < 0.001
Male sex		0.50 (0.34–0.76), *p* = 0.001
Background CV events	yes/no	2.39 (1.67–3.42), *p* < 0.001
Albumin	1 g/dL	0.64 (0.42–0.98), *p* = 0.04
Phosphate	1 mg/dL	1.22 (0.98–1.53), *p* = 0.08
Hemoglobin	1 g/dL	0.85 (0.76–0.94), *p* = 0.003

Variables out of the model: smoking (*p* = 0.22), diabetes (*p* = 0.19), GFR (*p* = 0.92), total cholesterol (*p* = 0.33), urinary protein (*p* = 0.34), phosphate (*p* = 0.12), ferritin (*p* = 0.90), iron (*p* = 0.85), BMI (*p* = 0.25), systolic blood pressure (*p* = 0.15), CRP (*p* = 0.34), and PTH (*p* = 0.24).

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
