# Peer review of "Serum Erythroferrone Levels Associate with Mortality and Cardiovascular Events in Hemodialysis and in CKD Patients: A Two Cohorts Study"

_jcm, 2019, doi:10.3390/jcm8040523_

Reviewer 1 Report

The manuscript entitled "Erythroferrone predicts mortality and cardiovascular events in hemodialysis" is an original and well written.

However, the quality could be improved.

Abstract: 192 words. Describes  properly the study

Introduction: well and properly stated on the ground of knowlegdment.

Material and Methods: Well study protocol described and the chart flow. Study outcome properly stated.  laboratory measurements the normal range of ERFE should be stated as orientation.

Statistically analysis: properly stated.

Results: Some issues should be clarified. Among CKD stages in the population selected is described asstages 2-5. In the results appears 3% of Stage 1. Also in the percentage of the patients ,the 26 patients are missing.

In the multivariate analysis, for cardiovascular events and all causes of mortality ERFE is weighted in a 4% in all analysis and it is difficult to understand the prediction in all causes of mortality. Tables 3, Supplementary S1, S2 and S3 the evidence of ERFE on these items is very low , less that Age, and much more than Diabetes status,  Phosphate and backgrounds CV events.

All these issues should be explained in order to improve the effect of ERFE.

Discussion properly stated.

Final: in accordance to the results the title does not reflect the manuscript findings.

Author Response

Point-to-point replies to the referee 1

Q1. Laboratory measurements the normal range of ERFE should be stated as orientation.

R1. The only two studies where ERFE was measured in healthy subjects are those by Honda et al. (see Ref. 12) and by Hanudel et al. (see Ref. 13). In these studies ERFE values were measured by Honda et al. in 16 controls (mean 0.5+ 0.06 ng/mL) and by Hanudel et al. in 161 healthy subjects [median 7.8 (4.7-13.2) ng/mL].This information is reported in the Discussion (pg. 10, L.6).

Q2: Among CKD stages in the population selected is described as stages 2-5. In the results appears 3% of Stage 1. Also in the percentage of the patients, the 26 patients are missing.

R2.  We are grateful to the referee for having spotted this mistake. Indeed, the CKD stages of our patients range from G1 to G5 (and not from G2 to G5). We amended this mistake throughout the manuscript.

Q3: In the multivariate analysis, for cardiovascular events and all causes of mortality ERFE is weighted in a 4% in all analysis and it is difficult to understand the prediction in all causes of mortality. Tables 3, Supplementary S1, S2 and S3 the evidence of ERFE on these items is very low, less that Age and much more than Diabetes status, Phosphate and backgrounds CV events.  All these issues should be explained in order to improve the effect of ERFE.

R3. We agree that the word “prediction” is inappropriate in our study and, for this reason, we substituted “prediction” with “association” throughout. The manuscript. Ours is not a prognostic study. We address an etiological hypothesis: is ERFE involved in the high risk of adverse clinical outcomes in CKD and HD patients? In observational studies, like ours, the nature of the link between a given exposure (ERFE) and study outcomes (death and CV events in our study) is tested by the multivariate Cox’s model where the factor in question is considered together with potential confounders. As pointed out by the referee, prediction is a different issue that demands prognostic analyses (risk discrimination, risk reclassification and risk calibration).

As to the question that the risk by ERFE is low, one should consider that the risk for death and CV events in the dialysis population is an inherently multifactorial problem and, in the multivariate scenario, the contribution of individual risk factors should be seen in relationship to other risk factors. As shown in Table 2, major risk factors like being 1-year older associated with an excess risk of the 4% (HR=1.04). An important risk factor peculiar to ESRD like a 1-mg higher serum phosphate entailed an excess risk of the 6% (HR=1.06) (Table 2). The excess risk by a 5 ng/ml higher ERFE is 4%, i.e. identical to the risk of being 1 year older, which we believe is not a trivial risk. In our analysis ERFE is 7th in rank order among the 18 factors we tested to explain the incidence rate of all-cause mortality and CV events among 18 risk factors specifically tested (Table 2).

Q4. In accordance to the results the title does not reflect the manuscript findings.

R4. To comply with the recommendation by the referee, we changed the title of our study which is now “Serum Erythroferrone levels associate with mortality and cardiovascular events in hemodialysis and in CKD patients: A two cohorts study”.

Reviewer 2 Report

In the manuscript entitled “Erythroferrone predicts mortality and cardiovascular events in hemodialysis and in CKD patients: A two cohorts study”, Spoto et al. demonstrated the relationship between dose of erythropoietin and serum erythroferrone (ERFE) levels and also showed that ERFE seems to be able to predict CVD events and mortality in hemodialysis patients and chronic kidney disease population. There remain, however, some concerns with regards to methodological issue, data presentation and interpretation.

Major comments

1.      Authors include the patients with stage 2-5 of CKD into one population; however, as you know, there should be huge difference in iron metabolism and erythropoiesis response between stage 2 and stage 5 of CKD. Hemoglobin levels, affected by kidney function, is also linked to serum ERFE levels. Authors need to subdivide CKD population and analysis for understanding more precisely about ERFE metabolism in CKD patients.

2.      There appears to be an overinterpretation of the results. For instance, it is claimed in the title and throughout the manuscript that ERFE predicts mortality and CVD events in hemodialysis and in CKD patients; however, hazard ratio of adjusted association between ERFE and mortality or CVD events is quite small and other factors such as diabetes, smoke, and background CV comorbidities, possess much bigger impact on mortality and the risk of CVD events. These data seems too weak to prove the relationship between ERFE and future incidence in this study and clinical importance of measuring serum ERFE level is questionable.

3.      There seems no information about what kind of erythropoietin stimulating agents (ESAs) these patients received during this experimental period. As Half-life span and efficacy of stimulating erythropoietic response appear to depend on the type of ESAs which presumably affect serum ERFE concentration.

4.      Authors should show whether these patients receive iron replacement therapy. If they received, authors should demonstrate the type of iron replacement therapy and analyze the impact of iron replacement therapy on ERFE and influence the risk of mortality and CVD events.

5.      Honda et al. already showed the efficacy of ESA on ERFE in 2016. Authors should show and deeply discuss the novelty of their research and data.

6.      As to analysis of adjusted association of ERFE onto mortality and CVD events in both hemodialysis patients and CKD patients, hazard ratio seems to be quite small. Authors should demonstrate other data supporting their hypothesis or proposal and deeply discuss how or why serum ERFE levels influence mortality and CVD event.

7.      Authors need to demonstrate serum hepcidin level on these patients and show the relationship between mortality, cardiovascular risk and serum hepcidin level and hemoglobin level on both hemodialysis and CKD patients.

8.      Authors should demonstrate the details of the cause of death and need to explain how and why ERFE impacts mortality.

9.      It has been demonstrated that ESA resistance is associated with increased mortality. An observational study in hemodialysis patients revealed that patients with low hemoglobin (<9.5 g/dL) and larger ESA dose exhibited a higher mortality risk. Authors should analyze and demonstrate the relationship between ESA resistance index(ERI) and serum ERFE levels. That data would be of help for understanding underlying mechanism and their hypothesis.

10.   Clinical importance of measuring serum ERFE level is questionable. Authors need to explain this issue.

Minor Comments

1.      Authors showed no gender differences in serum ERFE levels in Figure1. It seems not so important for your proposal and not necessary to spend one figure. Putting this data into supplementary should be enough.

Author Response

Point-to-point replies to the referee 2

Q1. Authors include the patients with stage 2-5 of CKD into one population; however, as you know, there should be huge difference in iron metabolism and erythropoiesis response between stage 2 and stage 5 of CKD. Hemoglobin levels, affected by kidney function, is also linked to serum ERFE levels. Authors need to subdivide CKD population and analysis for understanding more precisely about ERFE metabolism in CKD patients.

R1. The problem of confounding due to different degrees of renal dysfunction could be faced by stratification (as suggested by the referee) or by multiple data adjustment (the approach adopted by us). We choose this latter strategy to preserve the study power, i.e. to avoid the analytical problems that would be generated by the relatively small number of patients within discrete CKD strata. Table 3 reporting the backward multivariate analysis shows that the GFR largely remained out of the final model (P=0.92). In other words, the degree of renal dysfunction was not a confounder of the relationship between ERFE and the combined end-point including death and CV events.

Q2. There appears to be an over-interpretation of the results. For instance, it is claimed in the title and throughout the manuscript that ERFE predicts mortality and CVD events in hemodialysis and in CKD patients; however, hazard ratio of adjusted association between ERFE and mortality or CVD events is quite small and other factors such as diabetes, smoke, and background CV comorbidities, possess much bigger impact on mortality and the risk of CVD events. These data seems too weak to prove the relationship between ERFE and future incidence in this study and clinical importance of measuring serum ERFE level is questionable.

R2. We agree with the referee that prediction demands the assessment of discrimination, calibration and risk reclassification analysis whereas our study is etiological in nature. For this reason, to avoid misunderstanding between “etiology” and “prediction” we appropriately changed the wording across the paper (see Reply 3 to referee 1). In no part of the manuscript we suggest that ERFE should be measured in clinical practice. Ours is a hypothesis generating study which for the first time tests the relationship between serum ERFE and relevant clinical events such as death and cardiovascular events. We agree that the risk associated with ERFE levels is small and expand on this issue in reply to the Q6 (see below).

Q3. There seems no information about what kind of erythropoietin stimulating agents (ESAs) these patients received during this experimental period. As Half-life span and efficacy of stimulating erythropoietic response appear to depend on the type of ESAs which presumably affect serum ERFE concentration.

R3. ESA treated HD patients (n=785) mainly received intravenous EPO alpha (n=511, 65%) which was administered at the end of the HD session. Only a minority of ESA treated HD patients received subcutaneous Darbopoietin (n=274, 35%) administered at weekly or biweekly intervals. Blood sampling was performed the day after dialysis, i.e. when EPO levels are in a plateau phase. The subcutaneous Darbopoietin dose was administered at variable intervals before blood sampling. The type of ESA treatment (EPO versus DARBO) did not affect (P=0.57) the relationship between serum ERFE and ESA dosage indicating that in our population a 1-week dose of EPO induced the same increase in ERFE levels than a an equivalent weekly dose of DARBO. Thus, it is the ESA dose rather than the type of ESA the driving force raising serum ERFE. In this respect, our observational data are in line with experimental data by Honda (Ref.12) and Hanudel (Ref.13).

Q4. Authors should show whether these patients receive iron replacement therapy. If they received, authors should demonstrate the type of iron replacement therapy and analyze the impact of iron replacement therapy on ERFE and influence the risk of mortality and CVD events.

R4. All hemodialysis patients received intravenous iron (iron gluconate) and CKD patients not on dialysis oral iron supplements depending on iron deficiency along the recommendations of KDOQI guidelines. Intravenous iron therapy was profiled according to the clinical response. To prevent iron overload, intravenous iron was temporary stopped when ferritin levels were >500 ng/mL. In the methods (pg. 4, L.20-22 and pg. 5, L. 2-3)        we have now specified the general approach to iron treatment.

As described in the manuscript (pg. 8, L. 7-8) there were highly significant, albeit weak, associations between iron (r=-0.17, P<0.001) or ferritin levels (r=-0.12, P<0.001) and circulating ERFE. As shown in Table 2, in the multivariate analysis not only ERFE but also ferritin was related to the incident risk of death and CV events independently of other factors (P=0.003) while serum iron failed to significantly predict the same outcomes (P=0.09).

Q5. Honda et al. already showed the efficacy of ESA on ERFE in 2016. Authors should show and deeply discuss the novelty of their research and data.

R5. We have appropriately quoted the seminal observations by Honda (Ref 12) and Hanudel (Ref 13) in the introduction (pg. 3, L. 18-20) and in the discussion (pg. 10, L. 2-7). The novelty of our study is that it is the first describing an association between serum ERFE and mortality and cardiovascular events in hemodialysis patients and in pre-dialysis CKD patients. Ours is a study based on two cohorts including 1123 and 745 patients while the studies by Honda and Hanudel, which aimed solely at describing the relationship between ERFE and ESA treatment, where based on 59 (Honda study) and 51 CKD patients and 97 dialysis patients (Hanudel study), respectively. We focused on why studying the association of serum ERFE with clinical outcomes is a relevant research issue in the Introduction (the last 2 paragraphs) and transparently discussed (see Discussion) our interpretation of the main study findings.

Q6. As to analysis of adjusted association of ERFE onto mortality and CVD events in both hemodialysis patients and CKD patients, hazard ratio seems to be quite small. Authors should demonstrate other data supporting their hypothesis or proposal and deeply discuss how or why serum ERFE levels influence mortality and CVD event.

R6. We agree that the effect is not an impressive one. The risk for death and CV events in the dialysis population is an inherently multifactorial problem and in the multivariate scenario the contribution of individual risk factors should be seen in relationship to other risk factors. As shown in Table 2, major risk factors like being 1 -year older associated with an excess risk of the 4% (HR=1.04). An important risk factor peculiar to ESRD like a 1-mg higher serum phosphate entailed an excess risk of the 6% (HR=1.06). As specified in the reply to 3rd query by the first referee, ERFE is 7th risk factor in rank order explaining the incidence rate of all-cause mortality and CV events among the 18 risk factors simultaneously tested in the multivariate Cox model (Table 2). The excess risk by a 5 ng/ml higher ERFE is 4%, i.e. identical to the risk of being 1-year older, which we believe is not a trivial risk.

Q7. Authors need to demonstrate serum hepcidin level on these patients and show the relationship between mortality, cardiovascular risk and serum hepcidin level and hemoglobin level on both hemodialysis and CKD patients.

R7. Unfortunately, we could not measure serum hepcidin levels. Therefore, we recognize that this missing information is a limitation of our study (see Discussion pg.11, L. 12-13). Hepcidin is in the pathogenic pathway whereby ERFE impacts upon iron metabolism and anemia and, ultimately, on clinical outcomes. We can only say that adjustment for inflammation, the main driver of hepcidin in HD and CKD patients, and for ferritin, a main indicator of iron stores, had minimal, if any, effect on the risk for the combined outcome by ERFE in both the HD and the CKD cohorts included in this study.

Q8. Authors should demonstrate the details of the cause of death and need to explain how and why ERFE impacts mortality.

R8. We have now prepared a new Supplementary Table S1 detailing the causes of death in the two study populations. As recognized in the “study limitations” paragraph, ours is a hypothesis generating study. At pages 10-11, we dedicate 23 lines to discuss why ERFE is directly related with the incident risk of death and CV events. We believe that, like high BNP (which is per sé a protective factor), also high ERFE (in theory a protective factor as well) signals a response aimed at mitigating the high risk of ESKD and CKD in general. However, as discussed at page 11 L. 17-18 and 20-22, the issue can only be solved with experimental studies.

Q9. It has been demonstrated that ESA resistance is associated with increased mortality. An observational study in hemodialysis patients revealed that patients with low hemoglobin (<9.5 g/dL) and larger ESA dose exhibited a higher mortality risk. Authors should analyze and demonstrate the relationship between ESA resistance indexERI and serum ERFE levels. That data would be of help for understanding underlying mechanism and their hypothesis.

R9. ERI in large part reflects ESA dose because ESA dose is the numerator of the ratio defining the ERI. Like ESA dose, also the ERI was associated with ERFE levels (r=0.27, P<0.001) and, like in previous studies, ERI was associated with death and CV events (HR: 1.007, 95% 1.001-1.013, P=0.02). However, while ESA dose was an effect modifier of the relationship between serum ERFE and the study outcome, ERI did not modify such an association (P=0.79) indicating that it is the amount of ESA being administered rather than resistance to ESA the factor that modifies the relationship between ERFE and death and CV events.

Q10. Clinical importance of measuring serum ERFE level is questionable. Authors need to explain this issue.

R10. We entirely agree that by now there is no evidence supporting the adoption of the measurement of ERFE in clinical practice. In no part of our manuscript we made such a statement. Our hypothesis generating study suggests that ERFE is part of a complex counter-regulatory response to the high risk of ESKD and CKD (see reply to Q8).

Minor Comments

Q1. Authors showed no gender differences in serum ERFE levels in Figure1. It seems not so important for your proposal and not necessary to spend one figure. Putting this data into supplementary should be enough.

R1. As suggested by the referee, we have now included the Figure 1 as supplementary material (Supplementary Figure 1).

Round  2

Reviewer 2 Report

The second version of this paper has been improved and has got scientific value. I have no further concern in this paper.